# In-Situ Wireless Pressure Measurement Using Zero-Power Packaged Microwave Sensors

**DOI:** 10.3390/s19061263

**Published:** 2019-03-13

**Authors:** Julien Philippe, Maria Valeria De Paolis, Dominique Henry, Alexandre Rumeau, Antony Coustou, Patrick Pons, Hervé Aubert

**Affiliations:** Laboratoire d’Analyse et d’Architecture des Systèmes (LAAS), Université de Toulouse, Centre National de la Recherche Scientifique (CNRS), Institut National Polytechnique (INP), 7 avenue du Colonel Roche, 31031 Toulouse, France; maria-valeria.de-paolis@laas.fr (M.V.D.P.); dominique.henry@laas.fr (D.H.); alexandre.rumeau@laas.fr (A.R.); anthony.coustou@laas.fr (A.C.); patrick.pons@laas.fr (P.P.); herve.aubert@laas.fr (H.A.)

**Keywords:** chipless sensors, electromagnetic clutter mitigation, radar imagery technique, sensor packaging, wireless pressure sensors, zero-power sensors

## Abstract

This paper reports the indoor wireless measurement of pressure from zero-power (or passive) microwave (24 GHz) sensors. The sensors are packaged and allow the remote measurement of overpressure up to 2.1 bars. Their design, fabrication process and packaging are detailed. From the measurement of sensor scattering parameters, the outstanding sensitivity of 995 MHz/bar between 0.8 and 2.1 bars was achieved with the full-scale measurement range of 1.33 GHz. Moreover, the 3D radar imagery technique was applied for the remote interrogation of these sensors in electromagnetic reverberant environments. The full-scale dynamic range of 4.9 dB and the sensitivity of 4.9 dB/bar between 0.7 and 1.7 bars were achieved with radar detection in a highly reflective environment. These measurement results demonstrate for the first time the ability of the radar imagery technique to interrogate fully passive pressure sensors in electromagnetic reverberant environments.

## 1. Introduction

Wireless, zero-power (i.e., passive or without embedded battery) and chipless (i.e., without electronic circuit or integrated circuit) sensors constitute a very convenient solution for the measurement of any physical or chemical quantity, because their lifetime is unlimited and the remote monitoring of any physico-chemical quantities in harsh environments is made possible [1]. Literature has reported many developments on such passive and wireless sensors for measuring, e.g., temperature [2,3,4], humidity [5,6], stress [7,8], gas concentration [9,10,11] or pressure [12,13,14].

The state-of-the-art nature of passive pressure sensors is summarized in Table 1 and Table 2. Various approaches and technologies have been investigated, the use of inductor-capacitive (LC) cells [15], dielectric resonators (DR) [16], surface acoustic wave (SAW) devices [17], millimeter-wave identification (MMID) structures [18] and microwave planar resonators [12,19,20,21] (see Table 2). The sensing devices reported in the literature are often dedicated to pressure measurement at high temperatures (>400 °C), using structures based on sapphire [22] and ceramic [13,17,23,24,25,26,27] materials. The sensor performance is usually characterized by the available full-scale measurement range, the measurement accuracy, and/or the sensitivity, that is, the variation of an electromagnetic (EM) descriptor (e.g., the resonant frequency, S-parameters, at a fixed frequency [20,21] or the radar cross-section [12,20]) when a small change occurs in the physical or chemical quantity of interest.

Table 1 gives the performances at ambient temperature for the wireless pressure sensors reported in the literature. The full-scale range of the pressure, the full-scale measurement response and the sensor sensitivity are given for when the used EM descriptor is linearly dependent on the applied pressure. When the resonant frequency (>100 MHz) of a cavity was used as the EM descriptor, the highest full-scale range of reported fully passive pressure sensors was 1.1 GHz (see [28]).

Many industrial applications of wireless sensors require: (i) a reading range (or sensor-to-reader separation distance) of a few tens of meters in electromagnetic reflective or reverberant environments, and (ii) the use of a compact and portable reader of sensor data. However, most of the pressure sensors reported in the literature (see Table 1) are measured from non-portable and bulky equipment, e.g., network, spectrum or impedance analyzers. The notable exception is reported in Reference [27], where a portable and compact (17 cm × 17 cm × 8 cm) electronic phase-difference detector circuit was used for measuring the resonant frequency variation of a pressure sensor. Nevertheless, the detection range reported in Reference [27] was limited to only 17 mm.

The long-range detection and interrogation of wireless passive sensors in electromagnetic reverberant environments are very challenging [29,30]. The electromagnetic clutter eventually generated by metallic objects or structures such as walls, machines, pipes or shelves (as shown in Figure 1) affects the wireless monitoring of wireless zero-power sensors, due to the multi-path propagation effect, electromagnetic interferences, degradation of the signal-to-noise ratio and masking of the electromagnetic backscattering from sensors [31,32]. Few studies have discussed the wireless measurement of physical or chemical quantities in cluttered environments. In Reference [22], a waveguide was designed to prevent the impact of the undesirable electromagnetic reflections, but the proposed solution is suitable only for short reading ranges (up to few meters). The wireless measurement of a passive pressure sensor in an electromagnetic reflective environment is composed of three specific metallic objects in front of—or behind—the sensor and is reported in Reference [16]. No measurement results for different applied pressures were reported. To the best knowledge of the authors, no published study has been devoted to the in-situ wireless interrogation of passive pressure sensors in electromagnetic reverberant environments.

In this paper, the remote monitoring of zero-power pressure sensors in electromagnetic reverberant environments was investigated. A 3D radar imagery technique was used for the detection and remote interrogation of the sensor, and a generic method applicable to any wireless sensor is presented in order to derive the applied pressure from radar images. In addition, the full-scale range obtained from the measurement of the scattering parameters of the designed pressure sensors was found to be larger than the one reported (Table 1).

The design and packaging of the sensors is detailed in Section 2. The sensor characterization is reported in Section 3, and the wireless interrogation in electromagnetic reverberant environments is reported in Section 4. Finally, conclusions are drawn in Section 5.

## 2. Design and Fabrication of Packaged and Zero-Power Microwave Pressure Sensors

The geometry and topology of the passive microwave pressure sensor are reported in Figure 2 [20], while Table 3 and Table 4 give, respectively, the dimensions of the transducer and dielectric properties of the materials used. As can be observed, the sensing device is composed of a thin, flexible silicon membrane placed on the top of a circular cavity, within which the resonator operates.

The resonator is a microstrip half-wavelength coupled-line microwave (24 GHz) resonator designed to perform overpressure measurements up to 2.1 bars (see Figure 2). When a gradient of pressure is applied between the inside and outside of the cavity, it causes both mechanical and electromagnetic effects on the passive sensor. This pressure gradient leads to the deformation of the membrane and, consequently, causes a changing of the gap between the membrane and the planar resonator. Accordingly, the resonant frequency of the resonator is modified [21]. From the measurement of this modification, the pressure gradient can be derived, as will be shown in the next section. The transduction principle is explained in detail in Reference [21]. The performance of the EM sensor was actually predicted using the full-wave ANSYS HFSS simulation tool [33]. The simulation results (reported in Reference [21]), performed on a very similar structure, depicted agreement between the experimental and the simulation results. The manufacturing process was performed on a 500 µm thick borosilicate glass (B33) using the same protocol as reported in Reference [20]. A seed layer composed of 0.05 µm thick titanium (Ti) and 0.5 µm thick copper (Cu) is first deposited on the substrate. Next, the electrochemical growth of Cu allows an increase in the thickness, up to 9 µm of the resonator ports. The half-wavelength microstrip resonator is then patterned through the standard photolithography and etching process. However, the protection of the structure against oxidation is required, and the additional electro-less deposition of the thin gold (Au) layer is then performed to protect the resonator from further high-temperature fabrication steps. Next, the ground plane is implemented by the PVD deposition of a layer of 0.05 µm thick chromium (Cr) and 1.5 µm thick aluminum (Al) on the back-side of the substrate. The patterning of a 10 µm thick spin-coated low-losses microwave photoresist is used to build the 5.5 mm diameter cylindrical cavity. The last technological step consists of bonding the 100 µm thick high resistivity silicon (Si-HRS) membrane directly to the photoresist by thermo-compression, at a temperature of 100 °C. This curing allows the reticulation of the photoresist. This proposed technological process is fast and requires few fabrication steps and masks for photolithography. Furthermore, the manufacturing process ensures that the pressure sensor is sufficiently airtight for the proof-of-concept.

In order to perform the overpressure measurement, the microwave sensor is finally positioned inside an aluminum cavity with the dimensions 11.25 mm × 8.3 mm × 5.56 mm. It is worth emphasizing that the selection of the packaging dimensions is crucial in terms of avoiding the excitation of undesirable cavity resonant modes (see Figure 3 (i), (ii) and (iii)) and locating the resonant frequencies of cavity modes outside the radar bandwidth. The use of fifteen brass films with thicknesses of 80 µm, located under the sensing device, allows enhancement of the electrical contact between the resonator feeding lines and the SMA connectors. Furthermore, the air-tightness of the packaging is also crucial for measuring the performances of the sensing device. In order to achieve full air-tightness, long-term, conductive and hermetic silver grease (from chemtronics, Reference [34]) is applied at the surface between the cavity and its cover.

## 3. Microwave Characterization of the Packaged Pressure Sensor from S-Parameters

The sensitivity of the packaged sensor is investigated for various applied pressures up to 3.5 bars. The calibrated vector network analyzer (VNA) PNA-X N5247A from Agilent was used to measure the scattering parameters (S-parameters) of the sensor (Figure 3a–c) with the accuracy of ±13 MHz for the frequency and of ±0.05 dB for the S-parameter magnitude. For monitoring the applied pressure, the automated pressure calibrator (APC) module (series 600 from Mensor) connected to a compressed air tank, was used. The pressure was applied with a precision of ±2.5 mbars.

From S-parameter measurement, the following two descriptors were displayed as a function of the applied pressure: (i) the resonant frequency *f_res_* of the microwave resonator and, (ii) the magnitude *S*_11_ of the input reflection coefficient at *f_c_* = 23.8 GHz (Figure 4). The reproducibility of the measurement results was first checked from the multiple (10) connections and disconnections from the VNA of the packaged sensor. At atmospheric pressure, *f_res_* and *S*_11_ at *f_c_* = 23.8 GHz fluctuated at around 23.65 GHz and −20.87 dB, respectively, during the experiment, with variations of ±31 MHz and ±0.47 dB (that is, an error of ±0.13% on the *f_res_* and of ±2.3% on the *S*_11_ parameter). By taking into account the measurement errors of the calibration step, the total measurement accuracies on the resonant frequency and input reflection coefficient *S*_11_ were ±44 MHz (±0.19%) and ±0.52 dB (±2.5%), respectively.

The air-tightness of the cavity in photoresist was evaluated by the measurement of the resonant frequency variation, in time, when the transducer was placed in a vacuum chamber. A decrease of 7.54 MHz per hour was observed during almost 50 h. This air-tightness level was acceptable for the duration of our pressure monitoring (a couple of minutes). Figure 4 displays the variation of the resonant frequency *f_res_* and input reflection coefficient *S*_11_ at *f_c_* (=23.8 GHz) with respect to the applied pressure. It should be noted that no significant change of *f_res_* and *S*_11_ occurred for applied pressure above 2.1 bars. A possible interpretation of this measurement result is that the membrane is in contact with the planar resonator for pressure above 2.1 bars. As a consequence, the maximum applied pressure, which is detectable by the sensor, was about 2.1 bars.

Compared with our previous results [20], the resulting full-scale measurement range was slightly smaller, but the sensitivity to pressure variation in this range was significantly higher due to the improvement of the air-tight packaging. The measurement sensitivities were found to be 995 MHz/bar between 0.8 bars and 2.1 bars, and 4.2 dB/bar between 0.7 bars and 2.1 bars. Furthermore, the full-scale measurement range, in terms of resonant frequency variation, was found to be 1.33 GHz (5.6% of *f_res_*). This range is larger than the one reported in the literature for passive pressure sensors working above 100 MHz (see Table 1).

## 4. Wireless Pressure Measurement in Electromagnetic Reverberant Environments

The passive pressure sensors were intended to be functional in electromagnetic reverberant environments (see Figure 1). The detection and interrogation of sensors, performed here through the original 3D radar imagery technique described in Reference [35], used 24 GHz FM-CW radar (IMST DK-sR-1030e model, see Reference [36]). The chirp at the frequency *f_c_* = 23.8 GHz with a modulation bandwidth *B* = 2 GHz (±1 GHz around *f_c_*) was transmitted by means of a horn lens antenna (gain of 28 dBi and beam width of 6°), as shown in Figure 5. The theoretical depth resolution *d* of the radar was of *c/2B* (=7.5 cm), where *c* denotes the speed of light in a vacuum. The simultaneous remote reading of two passive targets is reported: the Target n°1 is the zero-power pressure sensor described in Section 2, while the Target n°2 is an antenna loaded by a variable microwave resistor that mimics a pressure variation (see, for example, Reference [35]). The scene, as shown in Figure 5, is composed of multiple metallic objects and structures, such as pipes, grids and large tanks. Parasitic echoes or clutter generated by the electromagnetic reflective environment may alter the signal backscattered by the two targets and, as a result, may render target detection a challenge. To analyze the impact of the electromagnetic clutter on the remote measurement of pressure variation, the Target n°1 was placed at the range of 6.5 m from the radar, just behind the tanks (see Figure 5a,b).

The sensing device of Target n°1 was: (1) terminated by a 50 Ω load on the first port and (2) connected on the second port to a horn antenna (gain of 20 dBi) through a delay line (characteristic impedance of 50 Ω) with the effective length of 0.6 m. This length allows placement of the sensing mode in a region offering a high signal-to-noise ratio. A 50 Ω load was chosen for impedance matching with cables and antenna. The pressure applied to the sensing device was generated by the APC module which was connected to a compressed-air tank. The applied pressure ranged from 0 to 2.5 bars. The Target n°2 was placed near the metallic tanks (see Figure 5b,c) and at 4.5 m from the radar. This target was a horn antenna (identical to that used for Target n°1), which was connected to a variable resistor through a delay line (characteristic impedance of 50 Ω). The effective length (1.2 m) of this line was chosen in order to place the radar echo associated with the sensing mode of Target n°2 in a region with low clutter (a shadow zone). The resistor was the input impedance of a two-port variable microwave attenuator (CVA35-30D commercialized by MCLI, see [37]). The available magnitude of the input reflection coefficient ranged from −19 dB to −1 dB, or, equivalently, the input impedance varied between 60 Ω and 530 Ω.

The radar beam scanning of the scene was performed from −15° to 10° in azimuth and from −10° to 10° in elevation. The obtained radar image of the scene is displayed in Figure 6 in the cut-plane *φ* = 2°. Among clutters, the antennas (red squares) and loading devices (red circles) associated with the two targets were detectable. These measurement results were obtained for the applied pressure of 2.4 bars (Target n°1) and a load impedance of 530 Ω (Target n°2). The location of the pressure sensor was assumed to be known. Consequently, the statistical analysis of radar echoes could be performed in a small region where the sensing mode was present and the eventual clutter was weak. The selection of this region, which was expected to mitigate the electromagnetic clutter, was performed from the computation of isosurfaces in radar images. An isosurface is defined as the set of image voxels having the same magnitude and is usually computed from the marching cubes method [38]. Figure 7 shows the superposition of many computed isosurfaces in a radar image obtained from multiple radar echo amplitudes between −50 dB and −30 dB. As expected, radar echo of Target n°1 increased when the applied pressure increased. From the computed isosurfaces, the small region in which, (1) the sensing mode is present and (2) the clutter is weak, can be easily identified and selected.

Next, inside the selected small region, the statistical estimator *e_Max_* of the pressure was defined as the largest echo level. A convenient linear variation of this estimator with respect to the applied pressure was obtained from 0.7 to 1.7 bars (see Figure 8a) with a sensitivity of 4.9 dB/bar. The statistical estimator was also computed for Target n°2, which was located 6.0 m from the radar (see Figure 8b). Again, the linear variation was observed when the reflection coefficient at the resistor input ranged from −7.9 to −1.0 dB. For reflection coefficients lower than −7.9 dB, the estimator was found to be less accurate and gave values lower than the noise level (−34.5 dB). Let $\varepsilon_{lin}$ be the deviation from the measurement results from the linear regression. This descriptor may be defined as follows:(1)$$\varepsilon_{lin}=\sqrt{\frac{1}{N}\overset{N}{\underset{n=0}{\sum }}{\left(e_{lin,n}-e_{meas,n}\right)}^{2}}$$
where *N* is the total number of measurement results, and $e_{lin,n}$ and $e_{meas,n}$ denote the results given by the linear model and by the radar data for the *n*-th measurement, respectively.

The deviation $\varepsilon_{lin}$ is of 0.11 dB. Moreover, the precision $\delta_{lin}$ of the linear model may be defined as follows:(2)$$\delta_{lin}=\frac{\varepsilon_{lin}}{\alpha \;\times \;\Delta P}$$
where ΔP denotes the full-scale range of the pressure sensor and *α* is the slope of the linear regression. The precision $\delta_{lin}$ of the linear model was only 2.2%.

The pressure sensor was designed for the wireless measurement of pressure between 0 and 2.5 bars in uncluttered environments. However, as can be observed from Figure 8a, the detection of applied pressure lower than 0.7 bars is not possible due to the presence of undesirable echoes (or clutter) in this particular environment. Advanced signal processing techniques (not reported here) could eventually be applied for increasing the signal-to-noise ratio and, as a result, for mitigating the undesirable clutter occurring for low applied pressures.

## 5. Conclusions

This paper reported a new packaged, portable, passive and wireless pressure sensor designed for in-situ pressure monitoring. The technological process allows the fabrication of highly sensitive pressure sensors, with sensitivities of 995 MHz/bar between 0.8 and 2.1 bars and 4.2 dB/bar between 0.7 and 2.1 bars. Compared with the state-of-the-art of passive and wireless pressure sensors (Table 1), this device exhibits the largest full-scale measurement range (1.33 GHz between 0.8 and 2.1 bars).

In electromagnetic reverberant environments, the proposed 3D radar imagery technique makes possible not only the simultaneous detection of several sensing nodes, but also the wireless measurement of the pressure with sensitivity of 4.9 dB/bar between 0.7 and 1.7 bars.

The results reported here confirm the benefits of 3D radar imagery techniques for performing wireless pressure measurements from zero-power sensors, even in electromagnetic reverberant environments. They demonstrate the ability of this technique to perform in situ wireless measurement of pressure in multi-path propagation channels.

## Figures and Tables

**Figure 1 sensors-19-01263-f001:**
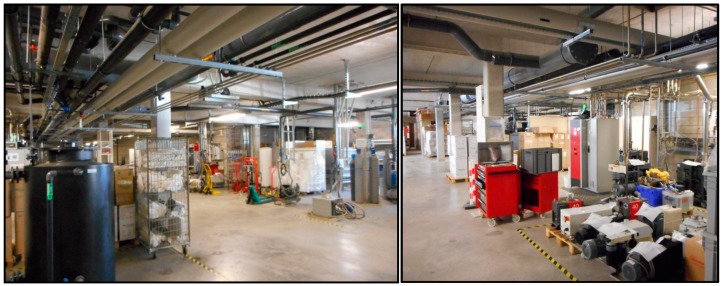
Industrial environments considered in this paper. The walls, metallic pipes and machines generate undesirable clutter that challenge the long-range detection and wireless reading of zero-power sensors for the monitoring of pressure.

**Figure 2 sensors-19-01263-f002:**
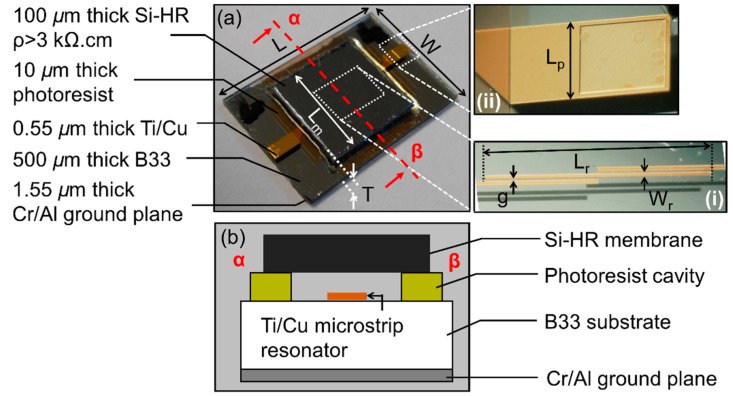
(**a**) Photos of the microwave (24 GHz) sensor after bonding with its high resistivity (Si-HR) membrane. *ρ* denotes the resistivity parameter. The insets (**i**) and (**ii**) show, respectively, the microstrip half-wavelength resonator below the membrane and the input or output port (9 µm thick Cu) of the planar resonator; (**b**) cross-sectional view of the different sensor layers [20].

**Figure 3 sensors-19-01263-f003:**
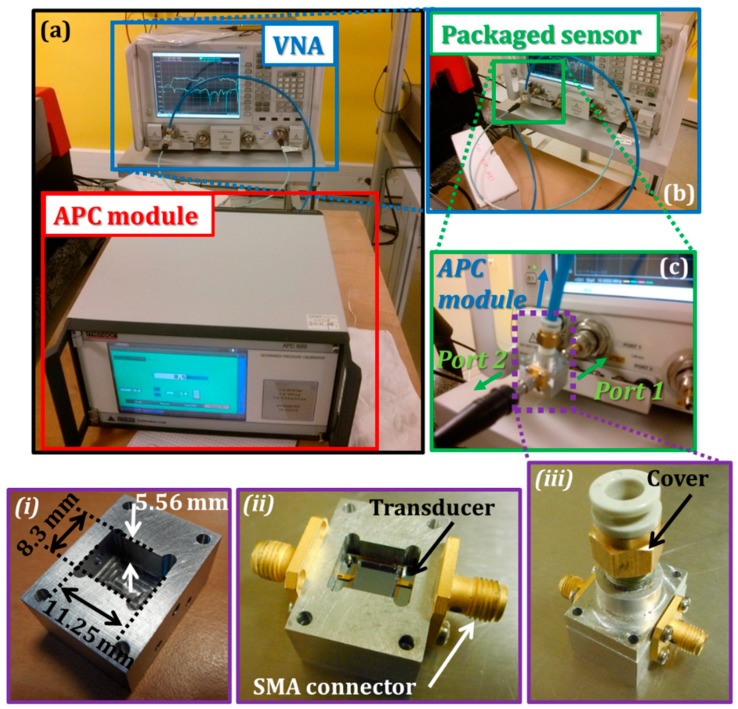
(**a**–**c**) Experimental setup for measuring the variation of the scattering parameters (S-parameters) when pressure is applied on the packaged sensor [21]. Insets (**i**), (**ii**) and (**iii**) represent photos of the sensor package. With its package, the dimensions of the pressure sensor are 36 mm × 24 mm × 42 mm.

**Figure 4 sensors-19-01263-f004:**
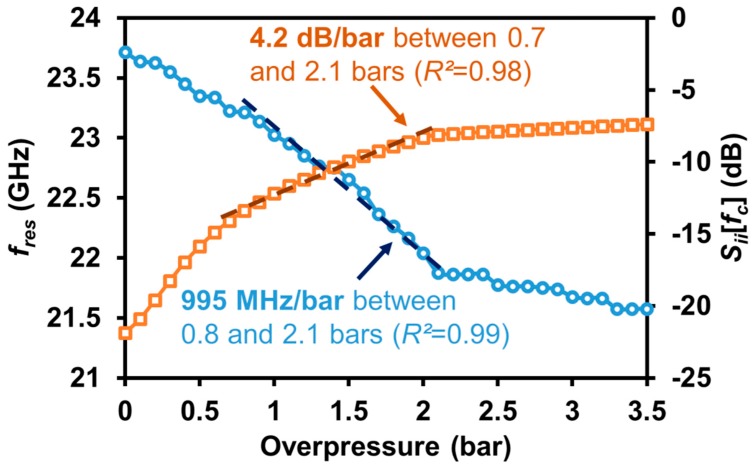
Variation of the measured resonant frequency *f_res_* and the input reflection coefficient *S*_11_ at the operating frequency *f_c_* (=23.8 GHz) as a function of the applied pressure. *R*^2^ denotes here the coefficient of determination of the linear regressions.

**Figure 5 sensors-19-01263-f005:**
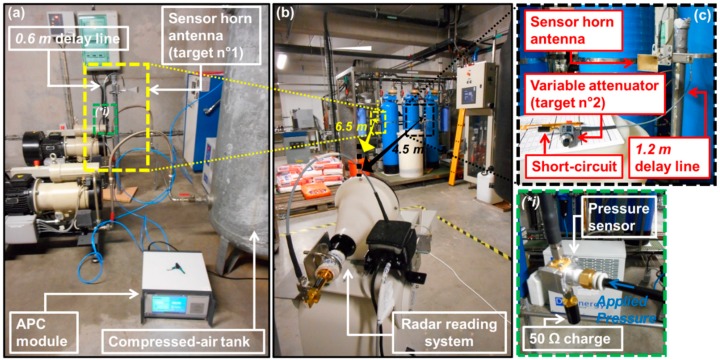
Wireless measurement setup in an electromagnetic reverberant environment. (**b**) indicates the position of Targets n°1 (pressure sensor) and Target n°2 (antenna loaded by a variable microwave resistor that mimics a pressure variation). (**a**) and inset (***i***) show the connection setup for Target n°1. (**c**) shows the connection setup for Target n°2.

**Figure 6 sensors-19-01263-f006:**
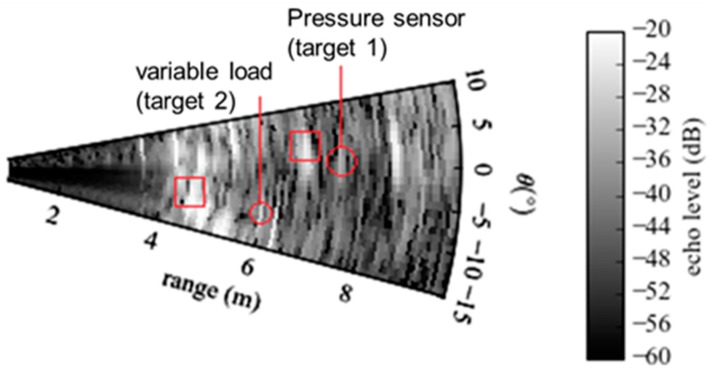
Radar image of the scene in the cut-plane *φ* = 2°. The scene includes two passive targets. (1) Target n°1: the zero-power pressure sensor with its horn antenna (the applied pressure is of 2.4 bars). (2) Target n°2: the horn antenna loaded by a variable microwave resistor (the resistance is of 530 Ω). The antennas are indicated by red squares while the loading devices (i.e., pressure sensor and variable resistor) are indicated by red circles.

**Figure 7 sensors-19-01263-f007:**
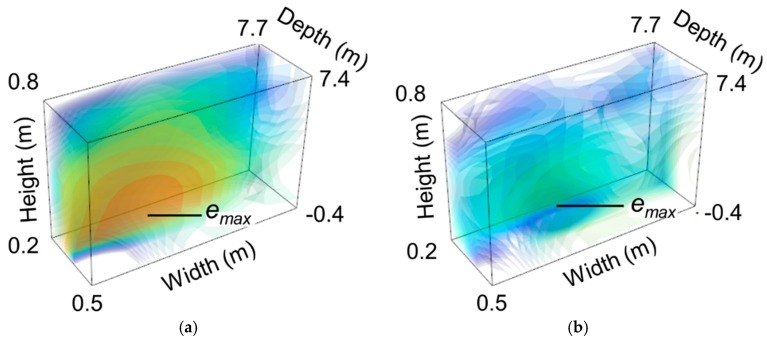
3D radar images of the zero-power pressure sensor using isosurfaces for applied pressures of (**a**) 1.9 bars and (**b**) 0 bar. Scale: −50 dB (blue) to −30 dB (red).

**Figure 8 sensors-19-01263-f008:**
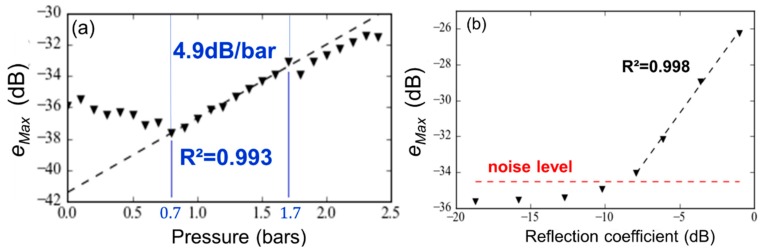
(**a**) Variation of the statistical estimator *e_Max_* with respect to the applied pressure (the sensing mode was located 7.5 m from the radar). A linear variation was obtained from 0.7 and 1.7 bars with the coefficient of determination *R*^2^ of 0.993 and slope (or measurement sensitivity) of 4.9 dB/bar; (**b**) Variation of the statistical estimator *e_Max_* as a function of the input reflection coefficient of the variable resistor (the sensing mode was located 6.0 m from the radar). The noise level at this position is around −34.5 dB.

**Table 1 sensors-19-01263-t001:** Characteristics of published wireless and passive pressure sensors—sensing performance.

Reference	Range of Applied Pressure ^a^	EM Estimator Value of Reference	Maximum Absolute Sensitivity (Relative) ^b^	Absolute Accuracy (Relative) ^c^	Absolute Full-Scale Response (Relative) ^b^	Equipment for Measuring the Device ^d^
[23]	0–3 bars	21.633 MHz	25.6 kHz/bar (0.12%/bar)	NR	75 kHz (0.35%)	SA Anritsu MS620J
[24]	0–2 bars	22.68 MHz	164 kHz/bar (0.72%/bar)	±19 mbars (0.95%)	0.3 MHz (1.3%)	IA HP9141A
[13]	0.7–2 bars	32.6 MHz at 400 °C	1.96 MHz/bar (6.0%/bar) at 400 °C	±50 mbars (3.8%)	2.6 MHz (8.0%) at 400 °C	NA Agilent E5061B
[28]	0.15–3 bars	39.3 GHz	370 MHz/bar (0.94%/bar)	NR	1.1 GHz (2.8%)	VNA
[12]	0–4 bars	−18.3 dBm	0.8 dBm/bar (4.4%/bar)	NR	4.8 dBm (26%)	FM-CW Radar
[17]	0–97 bars	434 MHz	8.4 kHz/bar (1.9 × 10^−3^%/bar)	NR	0.8 MHz (0.18%)	VNA Keysight E5061B
[19]	0–0.8 bars	9.61 GHz	16 MHz/bar (0.17%/bar)	NR	30 MHz (0.31%)	PNA Agilent E8362B
[22]	0.5–8 mbars	14.78 GHz	2.17 GHz/bar (15%/bar)	NR	15.4 MHz (0.10%)	NA Agilent E5071C
[16]	1–2 bars	20.53 GHz	455 MHz/bar (2.2%/bar)	NR	500 MHz (2.4%)	« Reader Device »
[25] ^e^	0–5.33 bars	11.75 GHz at 800 °C	35.88 MHz/bar (0.31%/bar) at 800 °C	NR	180 MHz (1.5%) at 800 °C	PNA-L Agilent 40 GHz
[26]	0–0.8 bars	18.94 MHz	0.344 MHz/bar (1.8%/bar)	NR	0.275 MHz (1.5%)	IA Agilent E4991A
[27]	1–2 bars	22.0 MHz	0.225 MHz/bar (1.0%/bar)	±12 mbars (1.2%)	0.25 MHz (1.1%)	Phase difference detector circuit
[20]	0.75–2.8 bars	23.65 GHz	440 MHz/bar (1.9%/bar)	±100 mbars (4.9%)	900 MHz (3.8%)	PNA-X Agilent N5247A
	1.2–2.4 bars	−21.9 dB	3.4 dB/bar (16%/bar)	NR	4.12 dB (19%)	
	0.5–2 bars	−43.9 dB	5.7 dB/bar (13%/bar)	±10 mbars (0.67%)	9.1 dB (21%)	FM-CW Radar
[21]	1–2 bars	22.95 GHz	620 MHz/bar (2.7%/bar)	±129 mbars (13%)	620 MHz (2.7%)	PNA-X Agilent N5247A
	1–2 bars	−6.92 dB	2.29 dB/bar (33%/bar)	NR	2.29 dB (33%)	
**This work**	**0.8–2.1 bars**	**23.65 GHz**	**995 MHz/bar (4.2%/bar)**	**±44 mbars (3.4%)**	**1.33 GHz (5.6%)**	**PNA-X Agilent N5247A**
	**0.7–2.1 bars**	**−21.9 dB**	**4.2 dB/bar (19%/bar)**	**±124 mbars (8.9%)**	**6.0 dB (27%)**	
	**0.7–1.7 bars**	**−38 dB ^f^**	**4.9 dB/bar (13%/bar)**	**±10 mbars (1.0%)**	**4.9 dB (13%)**	**FM-CW Radar**

^a^ The indicated ranges of applied pressure correspond to overpressures relative to atmospheric pressure. ^b^ The sensitivities and full-scale ranges are indicated by the absolute and relative values. The relative value is the ratio between the absolute value and the value of the electromagnetic (EM) descriptor at atmospheric pressure. NR—rot reported. ^c^ The measurement accuracies are indicated in absolute and in relative values with respect to the range of applied pressure. ^d^ The different acronyms corresponding to the equipment’s of measurement are defined here: Performance Network Analyzer (PNA); Vector Network Analyzer (VNA); Spectrum Analyzer (SA); Impedance Analyzer (IA); Network Analyzer (NA) and Frequency-Modulated Continuous-Wave (FM-CW). ^e^ In this reference, different forces were applied by using an alumina rod to characterize the pressure sensor. ^f^ The value of this EM descriptor is taken at the lowest pressure in the linear region. For the other cases, the value is given at atmospheric pressure.

**Table 2 sensors-19-01263-t002:** Characteristics of published wireless and passive pressure sensors—materials and dimensions.

Reference	Transducer	Materials *	Sensor’s Dimensions
[23]	LC resonator	Ceramic, Ag paste	56.4 mm × 40.7 mm × 0.56 mm
[24]	LC resonator	Ceramic, Ag ink	38 mm × 38 mm × 5.16 mm
[13]	LC resonator	Ceramic, Ag paste	36.2 mm × 36.2 mm × 0.57 mm
[12,28]	Microwave resonator	Glass, Al, Si	5.8 mm × 3.8 mm × 1.4 mm
[17]	SAW resonator	Ni, Cr, Steel, Quartz, Ceramic adhesive	177 mm² × ~7 mm
[19]	Microwave resonator	Si, Quartz, Al	5 mm × 4 mm × 0.3 mm
[22]	Capacitive sensor	Sapphire, Pt	10 mm × 10 mm × 0.32 mm
[16]	Dielectric resonator	Steel, DR coated by Ni and Au	1257 mm² × 5.5 mm
[25]	Evanescent-mode resonator	SiAlCN, PDC ceramic, Pt	134 mm² × 1.8 mm
[26]	LC resonator	Ceramic tape, Ag paste	26 mm × 26 mm × 0.5 mm
[27]	LC resonator	Alumina ceramic, Ag paste	3.3 mm × 3.3 mm × 0.48 mm
[20]	Microwave resonator	Glass, Si, photoresist	11.02 mm × 8.22 mm × 0.61 mm
[21]	Microwave resonator	Glass, Si, photoresist	11.02 mm × 8.22 mm × 0.61 mm
**This work**	**Microwave resonator**	**Glass, Si, photoresist**	**11.02 mm × 8.22 mm × 0.61 mm**

* Ag, Ni, Cr, Pt, Si and PDC stand respectively for silver, nickel, chromium, platinum, silicon and Polymer Derived Ceramic.

**Table 3 sensors-19-01263-t003:** Transducer dimensions (in mm).

L	W	T	L_p_	L_r_	W_r_	g	L_m_
11.02	8.22	0.61	0.85	3.5	0.075	0.02	6.0

**Table 4 sensors-19-01263-t004:** Dielectric parameters of sensor materials.

Material	Relative Permittivity	Loss Tangent
B33	4.6	9 × 10^−3^
Photoresist	3.5	2 × 10^−2^

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
