# Peer review of "In-Situ Wireless Pressure Measurement Using Zero-Power Packaged Microwave Sensors"

_sensors, 2019, doi:10.3390/s19061263_

Round 1

Reviewer 1 Report

This paper shows a novel approach for wireless measurement of air-pressure using a passive sensor and radar imaging.  If the authors make up following issues, it would be better.

1. For sensor development, repeatability of wireless measurement is the one of important indices.

I would like to recommend changing reader antenna locations (distances and angles from the target) and check whether the measured results are similar or not. If it is similar, what is an error range in terms of bar?

2. The authors designed a passive pressure sensor but did not explain sensing mechanism clearly.

If the conceptual idea of sensing mechanism is presented, it would be better.

3. Usually, EM sensor design is conducted with a simulation tool (e.g. CST, HFSS). However, the authors show neither their simulation results of the designed sensor nor performances matching with experimental results. If the author added this section, the paper would receive more attention from future readers.   

4. In Figure 9(a), eMax values are same (or very similar) at the pressure of 0.0 and 1.25 bars. I would recommend the authors should present how to distinguish these two cases.

Author Response

This paper shows a novel approach for wireless measurement of air-pressure using a passive sensor and radar imaging.  If the authors make up following issues, it would be better.

We thank the reviewer for his/her suggestions.

1. For sensor development, repeatability of wireless measurement is the one of important indices. I would like to recommend changing reader antenna locations (distances and angles from the target) and check whether the measured results are similar or not. If it is similar, what is an error range in terms of bar?

In the reported work, we use directive horn antennas with a 3dB-beamwidth of 20° and we assume that the sensor-target is located in the direction of antennas main beam. When the target is illuminated by the main lobe of these directive antennas, the wireless measurement does not depend on the angle from the sensor-target.  For achieving sensor detection and reading in almost all directions in space, so-called omnidirectional (or isoflux) antennas should be used, but at the expense of the reduction of reader-to-target separation distance (or detection range). Moreover, since the signal-to-noise ratio decreases as target-to-reader separation distance increases, the wireless measurement uncertainty is expected to grow as the distance increases. The impact of the reading range on the measurement accuracy, as well as the measurement repeatability, will be discussed by the authors elsewhere. 

2. The authors designed a passive pressure sensor but did not explain sensing mechanism clearly. If the conceptual idea of sensing mechanism is presented, it would be better.

The manuscript refers actually to [21] for the detailed description of the sensing mechanism. This mechanism is based on the well-known electromagnetic transduction. As the reported work is mainly devoted to the sensor reading in highly reflective environment, we decided to indicate the available authors’ reference [21] (and refs. therein) for describing this transduction. 

 3. Usually, EM sensor design is conducted with a simulation tool (e.g. CST, HFSS). However, the authors show neither their simulation results of the designed sensor nor performances matching with experimental results. If the author added this section, the paper would receive more attention from future readers.  

The performances of the EM sensor were actually predicted using the full-wave HFSS simulation tool. The simulation results were already reported in [21] on a very similar structure, and the experimental results were in a good agreement with the simulation results.

4. In Figure 9(a), eMax values are same (or very similar) at the pressure of 0.0 and 1.25 bars. I would recommend the authors should present how to distinguish these two cases.

The pressure sensor was designed for the wireless measurement of pressure between 0 bar to 2.5 bars in uncluttered environments. However, as it can be observed from Figure 9(a), the lowest  pressure derived  from radar echo analysis is of 1.25 bars due to the presence of undesirable echoes (or clutter). Advanced signal processing techniques (not reported here) could eventually be applied for increasing the signal-to-noise ratio and as a result, for mitigating this undesirable clutter occurring for low applied pressures.

Reviewer 2 Report

PAPER IS INTERESTING. ONLY MINOR CORRECTION AS FOLLOWS.

the microwave sensor is  positioned  inside an aluminium cavity of dimensions 11.25 mm x 8.3 mm x 5.56 mm. BASED ON ANY ASSUMPTION ?

It is worth underlying that the selection of the packaging dimensions is crucial to avoid the excitation of undesirable cavity  resonant modes (see Figure 3). HAVE YOU GOT IT USING TRIAL AND ERROR?

The use of some brass films with thickness of 80 μm located under the  sensing device allows enhancing the electrical contact between the resonator feeding lines and the SMA connectors. CAN YOU EXPLAIN HOW MANY BRASS FILMS USED? SOME IS LITTLE VAGUE..

In order to achieve a fully air-tightness, a long-term conductive and hermetic silver grease is applied at the surface between the cavity and its cover. CAN YOU ADD REFERENCES TO THE CONDUCTIVE/GREASE

Author Response

PAPER IS INTERESTING. ONLY MINOR CORRECTION AS FOLLOWS.
Authors’ reply: We thank the reviewer for his/her suggestions.

1. The microwave sensor is  positioned  inside an aluminium cavity of dimensions 11.25 mm x 8.3 mm x 5.56 mm. BASED ON ANY ASSUMPTION ?

Authors’ reply:  The sensor package was designed in order to locate the resonant frequencies of cavity modes outside the radar bandwidth.

2. It is worth underlying that the selection of the packaging dimensions is crucial to avoid the excitation of undesirable cavity  resonant modes (see Figure 3). HAVE YOU GOT IT USING TRIAL AND ERROR?

Authors’ reply: When designing the sensor package, we took into account the resonant frequencies of the cavity modes, the quality of the electrical connection between connectors and sensor’s ports, and the airtightness of the package. Some experiments were necessary to derive the final package dimensions, while electromagnetic simulations were performed for confirming that the predicted resonant frequencies of the cavity modes were located outside the radar bandwidth.

3. The use of some brass films with thickness of 80 μm located under the  sensing device allows enhancing the electrical contact between the resonator feeding lines and the SMA connectors. CAN YOU EXPLAIN HOW MANY BRASS FILMS USED? SOME IS LITTLE VAGUE..

Authors’ reply:  We used fifteen films of brass in order to guarantee a good electrical contact. This information is added in the revised manuscript.

4. In order to achieve a fully air-tightness, a long-term conductive and hermetic silver grease is applied at the surface between the cavity and its cover. CAN YOU ADD REFERENCES TO THE CONDUCTIVE/GREASE

Authors’ reply:  The conductive grease is a silver conductive grease from chemtronics: https://www.chemtronics.com/circuitworks-silver-conductive-grease. This reference is added in the revised paper.

Reviewer 3 Report

The paper describes a pressure sensor that can measure pressure with sensitivity of about 995 MHz/bar for a range of 2 bars or 4.9 dB/bar. The sensor is a passive, wireless, indoor device that uses 3D radar for detection of pressure. The sensor is useful. It may find other applications. However, there are several issues that need more clarification, especially for first time readers of the subject matter:

1-     In the abstract, it is mentioned that the full range is 4.9 dB, is this correct? It appears from Fig 5 to be about 10 dB. Please clarify.

2-     Clarify the need for the 50 ohms resistor termination (this may be clear to experts but not for students)

3-     Photoresist is used to build the cavity. Since photoresist is a soft material, even after baking or curing. Elaborate on the reliability of the sensor, especially when there is a pressure leak after certain time.

4-     Have the authors considered using a more reliable polymer, such as Polyimide instead of the photoresist? If so, can they elaborate why they did not use it, and if not why?

5-     The sensor uses radar images to read the pressure of the sensor. Elaborate on the advantage of using a radar system compared with using direct measurements using coaxial cable, as done with the testing, or using a simple RFID to detect the resonant frequency.

6-     Why using Si-HRS? what about using glass or other less expensive, high resistance material? 

Author Response

The paper describes a pressure sensor that can measure pressure with sensitivity of about 995 MHz/bar for a range of 2 bars or 4.9 dB/bar. The sensor is a passive, wireless, indoor device that uses 3D radar for detection of pressure. The sensor is useful. It may find other applications. However, there are several issues that need more clarification, especially for first time readers of the subject matter:

Authors’ reply: We thank the reviewer for his/her suggestions.

1. In the abstract, it is mentioned that the full range is 4.9 dB, is this correct? It appears from Fig 5 to be about 10 dB. Please clarify.

Authors’ reply: The full-scale measurement range of 4.9 dB, which is indicated in the abstract and reported in Figure 9a, is derived from the radar echo measured in the highly reflective environment, while the full scale measurement range of about 10dB given in Figure 5 is derived from S-parameters measurement of the pressure sensor. In the revised manuscript, the abstract has been modified for clarifying these two different full-scale measurement ranges.

2. Clarify the need for the 50 ohms resistor termination (this may be clear to experts but not for students)

Authors’ reply: In the reported work, the input impedance of the antennas, as well as the characteristic impedance of coaxial cables and microstrip lines, are of 50 Ohms. As a consequence, when the output port of the pressure sensor is loaded by the 50 Ohms impedance, the electromagnetic reflections from this port is suppressed. It can be observed (not shown here) that the full-scale measurement range of the sensor is significantly narrowed when this impedance matching is not performed.

3. Photoresist is used to build the cavity. Since photoresist is a soft material, even after baking or curing. Elaborate on the reliability of the sensor, especially when there is a pressure leak after certain time. Have the authors considered using a more reliable polymer, such as Polyimide instead of the photoresist? If so, can they elaborate why they did not use it, and if not why?

Authors’ reply: Polyimides could be actually more efficient to ensure a better airtightness than the used photoresist. However, their loss tangent tgd may be higher (for example, tgd = 0.0089 for HD 8820 material, while tgd = 0.0037 for the used photoresist).

4. The sensor uses radar images to read the pressure of the sensor. Elaborate on the advantage of using a radar system compared with using direct measurements using coaxial cable, as done with the testing, or using a simple RFID to detect the resonant frequency.

Authors’ reply: The targeted application is the mm-wave interrogation of multiple passive sensors distributed in electromagnetic reverberant environments and for distances of at least 10 meters. For a high number (> 10) of sensors, the wired solution using coaxial cables would be both expensive and cumbersome. Moreover, the interrogation of passive RFID tags would be very sensitive to multi-paths and collisions. Finally, the long range (> 10 meters) reading of multiple passive RFID tags in highly reflective environments would be technically very difficult to achieve in practice.

5. Why using Si-HRS? what about using glass or other less expensive, high resistance material? 

Authors’ reply: Unlike standard silicon, high resistivity silicon is suitable for our application because of its high resistivity (>3000 ohmNaN) and its dielectric permittivity (around 12). Such material was tested and approved in previous devices presented by the authors in [14]. Furthermore, thanks to its high machining facility, Si-HRS could allow a manufacturing of several sensors at the same time. Besides, high resistivity silicon was preferred here for its well-known mechanical properties and elasticity behaviour (i.e., the ratio between the deflection and applied pressure).